# Polymorphism, Expression, and Structure Analysis of a Key Gene ARNT in Sheep (*Ovis aries*)

**DOI:** 10.3390/biology11121795

**Published:** 2022-12-10

**Authors:** Xinyue Wang, Jingjing Bao, Yazhen Bi, Wenping Hu, Li Zhang

**Affiliations:** 1Institute of Animal Sciences, Chinese Academy of Agricultural Sciences, Beijing 100193, China; 2College of Animal Science and Technology, Qingdao Agriculture University, Qingdao 266109, China

**Keywords:** growth traits, ARNT, sheep, expression and function, polymorphism

## Abstract

**Simple Summary:**

Although the fact that ARNT may be related to sheep embryonic muscle growth and development has been clearly shown by our prior TMT proteomic study, its polymorphism, expression, and function remain unknown. Our goal was to thoroughly examine the association between the *ARNT* SNP loci and sheep growth traits, and elucidate the possible role of the *ARNT* gene in the formation and growth of sheep muscles. In this study, we evaluated the expression of the *ARNT* gene in the Hu, Chinese merino, and Gangba sheep, screened the polymorphism of the *ARNT* gene in the Hu and Ujimqin sheep populations, cloned the CDS sequence of the *ARNT* gene in the Hu sheep, and compared the CDS and protein sequences of the *ARNT* using bioinformatic analysis. The main results of our study showed an association between growth traits and sheep *ARNT* SNP loci, and a nonconservative missense point mutation G > C in the Hu and Ujimqin sheep, which is the primary factor that may regulate how the ARNT protein functions in the biology of sheep muscle growth and development. This work offers new and valuable insights into the mining of essential genes and identification of the molecular markers for sheep growth traits.

**Abstract:**

Growth traits are influential factors that significantly affects the development of the sheep industry. A previous TMT proteomic analysis found that a key protein in the HIF signaling pathway, ARNT, may influence embryonic skeletal muscle growth and development in sheep. The purpose of this study was to better understand the association between the polymorphisms of *ARNT* and growth traits of sheep, and the potential function of *ARNT*. Real-time qPCR (qRT-PCR) of *ARNT* was carried out to compare its expression in different developmental stages of the muscle tissues and primary myoblasts in the Hu, Chinese merino, and Gangba sheep. The genetic variance of *ARNT* was detected using the Illumina Ovine SNP 50 K and 600 K BeadChip in the Hu and Ujimqin sheep populations, respectively. The CDS sequence of the *ARNT* gene was cloned in the Hu sheep using PCR technology. Finally, bioinformatic analytical methods were applied to characterize the genes and their hypothetical protein products. The qRT-PCR results showed that the *ARNT* gene was expressed significantly in the Chinese merino embryo after 85 gestation days (D85) (*p* < 0.05). Additionally, after the sheep were born, the expression of *ARNT* was significant at the weaning stage of the Hu sheep (*p* < 0.01). However, there was no difference in the Gangba sheep.In addition, six SNP loci were screened using 50 K and 600 K BeadChip. We found a significant association between rs413597480 A > G and the Hu sheep weight at weaning and backfat thickness in the 5-month-old sheep (*p* < 0.05), and four SNP loci (rs162298018 G > C, rs159644025 G > A, rs421351865 G > A, and rs401758103 A > G) were also associated with growth traits in the Ujimqin sheep (*p* < 0.05). Interestingly, we found that a G > C mutation at 1948 bp in the cloned *ARNT* CDS sequence of the Hu sheep was the same locus mutation as rs162298018 G > C identified using the 600 K BeadChip, which resulted in a nonconservative missense point mutation, leading to a change from proline to alanine and altering the number of DNA, protein-binding sites, and the α-helix of the ARNT protein. There was a strong linkage disequilibrium between rs162298018 G > C and rs159644025 G > A, and the ARNT protein was conserved among the goat, Hu sheep, and Texel sheep. And, we propose that a putative molecular marker for growth and development in sheep may be the G > C mutation at 1948 bp in the CDS region of the *ARNT* gene. Our study systematically analyzed the expression, structure, and function of the *ARNT* gene and its encoded proteins in sheep. This provides a basis for future studies of the regulatory mechanisms of the *ARNT* gene.

## 1. Introduction

The enhancement of sheep growth traits has become a major social demand in China. Muscle growth and development are closely related to sheep growth traits. Some studies have found that the growth and development of skeletal muscle is a complex biological process, in which the varying signaling pathways and genes that regulate muscle fiber proliferation and differentiation are related to the oxygen content and energy metabolism [1], and the *HIF-1* gene has the potential function of regulating muscle cell growth and development via the Wnt signaling pathway [2,3,4].

HIF-1 consists of two subunits, including HIF-1α and HIF-1β (*ARNT*), which is not only an important transcription factor but also a transcription regulation factor of hypoxia stress in mammals, and HIF-1 can regulate the activity of multiple types of protein, such as cell surface receptors, transcription factors, cytoskeletal proteins, angiogenic growth factors, cytokines, survival factors, glucose transporters, and glycolytic enzymes [5,6,7]. HIF-1α and ARNT belong to the bHLH-PER-ARNT-SIM(PAS) protein family. Among them, the HIF-α exists in the cytoplasm, and the HIF-1α, HIF-2α, and HIF-3α are composed of it. Additionally, the expression of HIF-1α, HIF-2α, and HIF-3α is regulated by the oxygen content in the body. Under a normal oxygen content, these subunits are degraded, while under low oxygen content conditions, the transcription activity and stability of the three subunits are increased, and they combine with *ARNT* to form a dimer so as to regulate the transcription and expression of pre-emptive target genes. The three subunits have different functions under different oxygen conditions [8,9,10,11].

In contrast to HIF-α, *ARNT* is a binding partner of the aryl hydrocarbon receptor (Ahr) and HIF-α subunit. It is stable in the cytoplasm and nucleus under normal oxygen, hypoxic conditions have a nuclear localization signal that can enter the nucleus through an import/dependent pathway [12]. Meanwhile, some researchers reported that *ARNT* is only expressed in certain types of cells, and it has the same regulation function as the HIF-1α [13,14,15], but the function of *ARNT* in skeletal muscle cell proliferation and differentiation is not clear. Currently, it is believed that *ARNT* regulates glucose metabolism and myofibroblast proliferation and differentiation [16].

At the cellular level, *ARNT* can interact with the transcription factor Miz-1 of Myc, which can directly regulate several biological processes, such as the metabolism, cell cycle, and proliferation [17]. Deng et al. established a hypoxic rat model at the model biology level, showing that Microrna-103/107 participates in hypoxia-induced pulmonary artery smooth muscle cell proliferation by targeting *ARNT* [18]. The results of Deng et al. are similar to the results of Maktepe et al., and they all indicated that in *ARNT* knockout mice, the number of smooth muscle fibers was reduced [19]. Moreover, *ARNT* depletion inhibits muscle growth and function, and recent studies have shown that the knockdown of *ARNT* in primary muscle cell lines impinges on differentiation in vitro, whereas skeletal-muscle-specific *ARNT* depletion in young mice results in decreased whole-muscle *N1ICD* expression levels and limits muscle growth and regeneration, and the knockdown of HIF-1 in embryonic mice does not affect muscle growth and development but has a regulatory effect on postnatal skeletal muscle satellite cells [20]. Furthermore, *ARNT* may be involved in the regulation of muscle growth and development. The protein interacts with some pathways that are involved in muscle growth and development. It also forms a myofiber regulatory network that regulates sheep growth. In addition, our previous TMT proteomic analysis found that a key protein, *ARNT*, may influence growth traits in sheep. Thus, whatever the molecular function or regulatory mechanism of *ARNT* is, it is worth studying further [21,22].

In this study, the single-nucleotide polymorphisms (SNPs) of the sheep *ARNT* gene were discovered and genotyped, and the relationship between each SNP and the growth characteristics was examined. Using PCR techniques, the CDS cDNA sequence of the *ARNT* gene was cloned from the skeletal muscle of Hu sheep. The gene and potential protein products of the gene were studied using bioinformatics. qRT-PCR was used to assess the expression of the *ARNT* gene in two sheep muscle tissues and myoblasts at various developmental stages. This study lays the groundwork for future research investigating how sheep *ARNT* gene function and expression. Additionally, it serves as a foundation for additional research on the molecular processes controlling sheep meat output.

## 2. Materials and Methods

### 2.1. Expression of the ARNT Gene in Different Breeds of Sheep

#### 2.1.1. RT-PCR of the *ARNT* Gene

To study the expression of *ARNT*, primary myoblasts were obtained from the Hu and Gangba sheep. The *ARNT* gene was amplified and its expression was examined in Chinese Merino sheep that were 85 (D85), 105 (D105), and 135 (D135) gestational days old. Muscle samples from this period were critical for embryonic muscle growth and development and were identical to those from our previous proteomic study [23]. Additionally, to examine the expression of *ARNT*, skeletal muscle samples from the key developmental stages of Hu sheep, including newborn, 1.5-, 4-, and 6-month-old Hu sheep, as well as 4- and 6-month-old Gangba sheep, were chosen. After that, all the samples were placed in a cryogenic freezer (−80 °C) for preservation.

Using cDNA obtained from all the tissues of the Hu sheep, Chinese Merino sheep, and Gangba sheep and the primary myoblasts of the Hu sheep and Gangba sheep as templates, *ARNT*-RT primers, and reference gene (β-actin) primers, PCR amplification was conducted. Each 20 μL reaction consisted of 10 μL 2× Taq PCR Master Mix, 8 μL ddH_2_O, 0.5 μL forward primer, 0.5 μL reverse primer, and 1 μL cDNA. The PCR program began with denaturation at 95 °C for 5 min, followed by 25 cycles of denaturation at 95 °C for 30 s, annealing at 62 °C for 30 s, and extension at 72 °C for 5 s, followed by a final extension at 72 °C for 8 min. The amplified products were analyzed by agarose gel electrophoresis. The quality of the cDNA obtained from reverse transcription was amplified using the primers of the reference gene, β-actin, to verify that it could be used for the subsequent experiments.

#### 2.1.2. RT-qPCR of the ARNT Gene in Sheep

Using all the cDNA as templates, RT-qPCR was conducted. Each 20 μL reaction contained 10 μL SYBR Premix Ex Taq II, 6.4 μL RNase-Free ddH_2_O, 0.8 μL forward primer, 0.8 μL reverse primer, and 2 μL cDNA. The RT-qPCR reaction started with denaturation at 95 °C for 5 s, followed by 40 cycles of denaturation at 95 °C for 5 s and annealing at 60 °C for 30 s. Each sample was run in triplicate, and β-actin was used as the reference gene.

#### 2.1.3. Statistical Analysis

The relative expression of each target gene was calculated using the 2^−△△Ct^ method. The data were analyzed with the statistical software SPSS 26.0, and the statistical data were reported as the mean ± the standard deviation. GraphPad Prism 8 with the one-way ANOVA statistical method was used to detect differences in expression between the different tissues, and Student’s *t*-test was used to compare the differences in expression among the same tissues. A value of *p* < 0.05 indicated a significant difference. All the primers are shown in Appendix A.

### 2.2. Sequence Polymorphism of the Sheep ARNT Gene

#### 2.2.1. DNA Extraction and Detection

DNA was collected from the blood using a blood genomic DNA extraction kit as per the manufacturer’s instructions (Omega Co., LTD, Beijing, China). Thermo Scientific’s Nano Drop 2000 spectrophotometer was used to measure the purity of the DNA and the concentration, and 1.5% agarose gel electrophoresis was used to verify the integrity of the genomic DNA. If the ratio of 260 to 280 nanometers was between 1.8 and 2.0 and the concentration was greater than 50 ng/mL, the sample was deemed acceptable.

#### 2.2.2. Quality Control and Genotype Extraction

The genotypes of the *ARNT* gene SNP were extracted from the Hu and Ujimqin sheep through the Illumina Ovine SNP 50 K and 600 K BeadChip using PLINK 19.0 software. The original data were also subjected to a quality check using PLINK 19.0 software, with the following standards: the minimal allele frequency, individual detection rate, and SNP detection rate were all more than 0.05, 0.90, and 0.90, respectively [24].

#### 2.2.3. Statistical Analysis

The genetic parameter formula was used to compute the polymorphism information content (PIC), homozygosity (Ho), the effective number of alleles (Ne), and heterozygosity (He) in Microsoft Excel 2019. Using SPSS 26.0, a chi-square test was performed for the genotype and gene frequencies of each SNP locus in the Hu and Ujimqin sheep, and the relationship between the genotype frequencies and growth characteristics was examined.

### 2.3. Linkage Disequilibrium of Associated SNP Markers in the ARNT Gene

The genotype and locus data for the five SNP loci (rs162298018 G > C, rs159644025 G > A, rs421351865 G > A, rs401758103 A > G, and rs427535938 G > A) of the *ARNT* gene were combined in two text files using the PLINK 19.0 software and 600 K BeadChip, respectively. Then, the Haploview software package (Broad Institute, Cambridge, MA, USA) [25] was used to estimate and plot the pairwise linkage disequilibrium (LD) measures.

### 2.4. Association between SNP Loci of the ARNT and Growth Traits

#### 2.4.1. Materials

We selected 336 Ujimqin sheep from two farms in Inner Mongolia and 3024 Hu sheep from Gansu province, China. From each sheep, we collected 10 mL of jugular vein blood and stored it at 4 °C after anticoagulation with EDTA. Additionally, the main growth traits of the Ujimqin sheep and Hu sheep were measured. The growth traits of the Ujimqin sheep mainly included body weight (BW), chest circumference (CC), body length (BL), tube circumference (TC), and body height (BH) at 4-month-old, and the BW, CC, BL, TC, GH, chest width (CW), and chest depth (CD) at 6-month-old. The growth traits of the Hu sheep mainly included the birth weight, weaning weight, and weight at 2-month-old. The BW, BH, BL, CC, TC, CW, and CD of the Hu sheep at 3-, 4-, 5-, and 6-month-old, respectively. Moreover, the cross height (CH) and chest width (CW) of the Hu sheep were measured at 3- and 4-month-old, and the backfat thickness (BFT) and eye muscle area (EMA) of the Hu sheep were measured at 5- and 6-month-old.

#### 2.4.2. Statistical Analysis

Using the general linear model (GLM) in R, the phenotypic data of the growth traits of the sheep were adjusted, and the model incorporated effects related to the sex, birth year, birth season, and age [26]. The detailed R run code is shown in Appendix A. The SPSS 26.0 Kolmogorov–Smirnov test (KS) of the corrected phenotypes revealed that the corrected data complied with the standards of a normal distribution and each observation was independent. Then, using the GLM in SPSS-26.0 software, the association analysis of the genotype and phenotype data was performed. The results were expressed as the mean ± standard error (mean ± SE), and multiple comparisons were performed using Duncan’s method, with *p* < 0.05 denoting a significant difference. The relevance analysis model used the following formula: Y_ijklm_ = u + G_i_ + D_j_ + M_k_ + Y_l_ + S_m_ + e_ijklm_, where Y_ijklm_ is the phenotypic value of the growth traits, μ is the population mean, G_i_ is the genotype effect, D_j_ is the day–age effect, M_k_ is the gender effect, Y_l_ is the year of birth, S_m_ is the seasonal effect, and e_ijklm_ is the random residual impact of each observation.

### 2.5. cDNA Cloning and Differential Expression Analysis of the ARNT Gene in Sheep

#### 2.5.1. Materials

Skeletal muscle samples were acquired from Hu sheep of a gestational age of 72 days, and all the samples were transferred to a cryogenic refrigerator (−80 °C) for storage.

#### 2.5.2. cDNA Cloning of the *ARNT* Gene

Extraction of total RNA from the sample

Samples from the Hu sheep of a gestational age of 72 days were ground in liquid nitrogen, and then the total RNA was extracted with the Total RNA Extraction Kit for Animal Tissues (DP431). The RNA integrity was assessed by 1% agarose gel electrophoresis, and the RNA concentration and purity were measured by nanodrop spectrophotometry. Only RNA samples displaying a 260 nm/280 nm ratio between 1.8 and 2 were used in this study. The product was stored in a −80 °C cryogenic freezer until further use.

2.Coding sequence (CDS) cloning of the *ARNT* gene

The total RNA of the embryonic skeletal muscle of the Hu sheep was subjected to RT-PCR using the Prime Script™ RT reagent kit. The reverse transcription products were stored at −20 °C. According to the predicted *ARNT* gene sequences (NCBI reference sequences: NM_001287465.1), the amplification primers of *ARNT* were designed with Premier 5.0 software (Appendix A).

The cDNA from the embryonic skeletal muscle of the Hu sheep obtained by RT-PCR, and those cDNA were used as templates to clone the *ARNT* gene using the TaKaRa LA Taq^®^ with GC Buffer (RR02AG) enzyme system. Each 20 μL reaction contained 1 μL cDNA, 0.5 μL forward primer (Appendix A), 0.5 μL reverse primer, 10 μL 2× Taq PCR Master Mix, and 8 μL ddH2O. The PCR program started with denaturation at 95 °C for 5 min, followed by 40 cycles of denaturation at 95 °C for 30 s, annealing at 62 °C for 30 s, and extension at 72 °C for 5 s, followed by a final extension at 72 °C for 3 min. The amplified products were detected by 1.0% agarose gel electrophoresis. The target fragments were recovered using the TaKaRa MiniBEST Agarose Gel DNA Extraction Kit Ver.4.0 (9762), ligated to a pMD18-T (TaKaRa) vector, and transformed into DH5α competent cells. Ampicillin-resistant colonies were screened, and the DNA fragments were sent to Sangon Biotech Co., Ltd. (Shanghai, China) for sequencing.

### 2.6. Bioinformatic Analysis of ARNT CDS

The Hu sheep *ARNT* CDS clone sequence and Texel sheep (*Ovis aries*, GenBank accession: NM_001287465.1) *ARNT* CDS sequence were compared using ApE. The Extasy translate tool “https://web.expasy.org/translate/ (accessed on 22 October 2022)” was used to predict the amino acid sequence. ProtParam “http://web.expasy.org/protparam/ (accessed on 22 October 2022)” was used to predict the physical parameters of each protein, such as the molecular weight. The hydrophilicity and hydrophobicity of the ARNT of the Hu sheep were analyzed using ProtScale “https://web.expasy.org/protscale/ (accessed on 22 October 2022)”. SignalP-6.0 “https://services.healthtech.dtu.dk/service.php?SignalP-6.0 (accessed on 22 October 2022)” was used to predict the signal peptides [27]. TMHMM “https://services.healthtech.dtu.dk/service.php?TMHMM-2.0 (accessed on 22 October 2022)” was used to predict the transmembrane helices. WoLF PSORT “https://wolfpsort.hgc.jp/ (accessed on 22 October 2022)” was used to predict the protein subcellular localization [28]. The prediction of the secondary structure of each protein and its variants was carried out using PredictProtein “https://www.predictprotein.org/ (accessed on 25 October 2022)”. SWISS-MODEL “https://swissmodel.expasy.org/interactive (accessed on 02 November 2022)” was used to predict the protein tertiary structure [29], and a phylogenetic tree was constructed using the neighbor-joining method in MEGA11 software [30,31,32].

## 3. Results

### 3.1. Differential Expression Analysis of the ARNT Gene in the Skeletal Muscle of Sheep at Different Development Stages

The *ARNT* gene expression in the muscle tissue and primary myoblasts of the different breeds of sheep was studied at different developmental stages by RT-qPCR (Figure 1). At the embryonic development phase, we analyzed the *ARNT* expression in the primary myoblasts of the Hu and Gangba sheep of the embryonic age of 72 days and the muscle tissue of the Chinese merino sheep at 85 (D85), 105 (D105), and 135 (D135) gestational days. The target gene expression analysis results showed no significant difference between the Hu and Gangba sheep primary myoblasts (Figure 1b). The expression level in the Chinese merino sheep fetal muscle at D85 was higher than that at D105 and D135. At D135, it was the lowest, and it was significantly lower than that at D85 (*p* < 0.05) (Figure 1a). Using the muscles of 4- and 6-month-old Gangba sheep and newborn, weaning, and 4- and 6-month-old Hu sheep, we analyzed the expression of *ARNT* at the postnatal developmental stage. All the results indicated that the *ARNT* expression was similar at both developmental stages in the Gangba sheep. (Figure 1c). Meanwhile, in the Hu sheep, not only was *ARNT* highly expressed in the muscle tissue during weaning, but there were also significant differences among the stages of weaning, 4- and 6-month-old (*p* < 0.01). (Figure 1d). At the same time, we found that the Gangba sheep had a higher expression of *ARNT* than the Hu sheep at 4- and 6-month-old.

### 3.2. The quality Inspection Results of the Illumina Ovine SNP 50 K and 600 K BeadChip

For the 600 K BeadChip, 606,006 SNPs covering the Ujimqin sheep were detected. Finally, 522,419 stable SNP marker loci and 333 individuals were obtained, and genotypes were extracted from the microarray data after the quality control stage. In the same way, 53,499 SNPs covering the 50 K BeadChip were detected. A total of 47,806 stable SNP marker loci and 2982 individuals were obtained, and genotypes were extracted from the microarray data after the quality control stage.

### 3.3. SNP Identification and Polymorphism Analysis Based on the Illumina Ovine SNP 50 K and 600 K BeadChip

Based on the Illumina Ovine SNP 50 K and 600 K BeadChip data derived from our laboratory, all the SNPs of the *ARNT* gene were scanned and checked, and ten SNP loci of the Hu sheep *ARNT* were identified, among which one SNP loci, rs413597480 A > G, for the 50 K BeadChip, and five SNP loci of the Ujimqin sheep, rs162298018 G > C, rs159644025 G > A, rs421351865 G >A, rs401758103 A > G, and rs427535938 G > A, for the 600 K BeadChip were successfully extracted. Among them, rs162298018 G > C in the Ujimqin sheep was in exon, and the rest of the nine sites in the two breeds of sheep were in introns. Only one exonic SNP was a case of G/C transversions (Table 1).

The polymorphism analysis of the *ARNT* SNPs showed that the allele frequency of all the SNP loci was greater than 10%, the dominant genotype of rs413597480 A > G in the Hu sheep was AA, and the dominant genotypes of rs162298018 G > C, rs159644025 G > A, rs421351865 G > A, rs401758103 A > G, and rs427535938 G > A in the Ujimqin sheep were GC, GG, GG, AA, and GG, respectively. The dominant allele of rs159644025 G > A, rs421351865 G >A, and rs427535938 G > A in the Ujimqin sheep was G, that of rs401758103 A > G was A, and the allele frequencies of rs413597480 A > G in the Hu sheep and rs162298018 G > C in the Ujimqin sheep were equal. Meanwhile, there were four SNP loci with low polymorphism (0.25 < *PIC* < 0.5), such as rs413597480 A > G in the Hu sheep and rs421351865 G > A, rs401758103 A > G, and rs427535938 G > A in the Ujimqin sheep. Moreover, rs162298018 G > C and rs159644025 G > A were moderately polymorphic (0.25 < *PIC* < 0.5) in the Ujimqin sheep. Additionally, all the SNP loci of *ARNT* were in Hardy–Weinberg equilibrium (*p* > 0.05) (Table 2).

### 3.4. Linkage Disequilibrium of the Associated SNP Markers in the ARNT Gene

From Figure 2, it can be seen that there was a strong linkage disequilibrium between rs162298018 G > C and rs159644025 G > A (Figure 2a). A haplotype domain (Figure 2b) was detected between rs159644025 G > A, rs421351865 G > A, rs401758103 A > G, and rs427535938 G > A. All the SNP loci in the haplotype domain were strongly in linkage disequilibrium. Additionally, five haplotypes existed in this haplotype domain, in which the highest frequency of the GGAG haplotype was 0.510.

### 3.5. Association between the SNP Loci of ARNT and Growth Traits in the Hu and Ujimqin Sheep

To demonstrate the association between the SNPs and the growth traits of the sheep more intuitively, we carried out an association analysis between the number of individuals of each genotype and the growth traits. The result of the growth traits is expressed as the average with standard deviation, and there were significant differences in different developmental stages between the different genotypes of *ARNT* in the Hu sheep. The body weight (BW) at the weaning stage was strongly associated with rs413597480 A > G. This mutant locus in the Hu sheep also showed a strong association with the chest circumference (CC) and backfat thickness (BFT) at 5-month-old, as well as the chest width (CW) at 6-month-old (*p* < 0.05) (Table 3, Table 4 and Table 5). In the Ujimqin sheep, no significant difference in the growth traits between the different genotypes of rs427535938 G > A was found (*p* > 0.05), and only four SNP loci were associated with the growth traits. The SNP variant rs162298018 G > C is located in the exon of *ARNT*, which is a G/C transversion and associated with the BW and CC at 4-month-old and the tube circumference (TC) at 6-month-old, and the Ujimqin sheep with the GG genotype had a greater BW and CC in the case of the 4-month-old sheep and greater TC in the case of the 6-month-old sheep compared to those with the CC and GC genotypes, respectively (*p* < 0.05). Meanwhile, the CC genotype of the rs162298018 G > C SNP variant was associated with the 6-month-old CC. In the Ujimqin sheep, the second *ARNT* locus was rs159644025 G > A, which was only associated with the 6-month-old growth traits, such as the BW, BH, BL, CC, and CW, and the AA genotype showed the highest growth trait values among all the genotypes. The SNP rs401758103 A > G was the third SNP loci of *ARNT* in the Ujimqin sheep. This SNP was associated with the BW at the 4-month-old stage and the TC at the 6-month-old stage, and the AA genotype of rs401758103 A > G in the Ujimqin sheep had a larger BW and TC. The last SNP mutation of *ARNT* in the Ujimqin sheep was rs421351865 G > A, and this SNP was associated with both the BW at the 4-month-old stage and TC at the 6-month-old stage, and the GG genotype of rs421351865 G > A showed a larger BW and TC (Table 6 and Table 7).

### 3.6. CDS Cloning of the ARNT Gene in Hu Sheep

To further analyze whether there were SNP loci in the CDS region of the *ARNT* gene in Hu sheep, we used the RT-PCR method to clone the CDS sequence of *ARNT*, and the product of the *ARNT* gene from the Hu sheep was 2373 bp (Figure 3a). The *ARNT* gene from Hu sheep was 99% similar to that of the Texel sheep (*Ovis aries*, GenBank accession: NM_001287465.1), with two base changes in the Hu sheep, and 99% similar to the goat gene (*Capra hircus*, GenBank accession: XM_018046044.1). Overall, the high level of similarity among the Hu sheep clone CDS, Texel sheep and goat indicated that this gene was very conservative. Meanwhile, a comparison of the CDS regions of the *ARNT* gene between the Hu and Texel sheep showed that the A > G variant was a synonymous mutation at the position of 645 bp, and the G > C of the *ARNT* CDS sequence results in a nonconservative missense point mutation, leading to a change from proline to alanine at the position of 1948 bp (Figure 3b). Interestingly, we discovered that the G > C mutation at 1948 bp in the cloned *ARNT* CDS sequence of the Hu sheep was the same as the rs162298018 G > C in Ujimqin sheep.

### 3.7. Bioinformatic Analysis of the ARNT Protein

#### 3.7.1. Physicochemical Properties of the ARNT Protein

The ARNT amino acid sequence of the Hu sheep was predicted, and the Texel sheep ARNT protein sequence was downloaded from NCBI. Additionally, ProtParam revealed that the Hu sheep ARNT protein was formed of 790 amino acids, the molecular weight of the ARNT protein was 86.64 kDa, the theoretical pI was 6.15, and the instability index was computed to be 53.5. Meanwhile, the ARNT protein of the Texel sheep was formed of 790 amino acids, the molecular weight of the ARNT protein was 86.66 kDa, the theoretical pI was same as that of the Hu sheep, and the instability index was computed to be 54.3. Overall, ARNT was an unstable protein in Hu or Texel sheep.

As shown in Figure 3, the ARNT proteins of Hu and Texel sheep had no signal peptides (Figure 4a), and this type of protein was a hydropathicity protein. The N-terminus of the ARNT protein and the C-terminus had many hydrophilic regions, among which arginine, at position 100, was the most hydrophilic (the score was −3.1). Alanine, at position 339, was the most hydrophobic (the highest score was 2.056), and hydrophobic and hydrophilic amino acids were also evenly distributed in the mature peptide region (Figure 4b). Moreover, there were no transmembrane helices in the ARNT protein, and all the amino acids were outside the membrane (Figure 4c). The results of the subcellular localization showed that ARNT was a nucleus protein.

#### 3.7.2. Advanced Structural Prediction and Phylogenetic Analysis of the ARNT Protein

The ARNT secondary structures of the Hu sheep and Texel sheep were predicted (Figure 4). The amino acid sequence of ARNT changed from proline to alanine at position 649 in the Hu and Texel sheep (*Ovis aries*, GenBank accession: NM_001287465.1). The predicted secondary structures showed that there were ten protein-binding regions in the Hu sheep ARNT protein, including eight low relative B-value (RI: 0–33) protein-binding regions and two intermediate relative B-value (RI: 34–66) regions, and the Texel sheep ARNT protein had one less low relative B-value protein-binding region (760) and one less intermediate relative B-value protein-binding region (762) than the Hu sheep. The Hu sheep had twenty-three low relative B-value (RI: 0–33) DNA-binding regions, and the Texel sheep had twenty-four low relative B-value (RI: 0–33) DNA-binding regions. While the Hu sheep ARNT protein had twelve intermediate and two high relative B-value (RI: 66–100) DNA-binding regions, they had three more intermediate (703, 707, 711) and one more high relative B-value DNA-binding region (672) than the Texel sheep. Last but not least, the results for the RNA-binding region showed that the same number of high relative B-value RNA-binding regions can be found in Hu and Texel sheep. While the low RNA-binding sites of the Texel sheep were two more in number than those of the Hu sheep (624,734), and the intermediate RNA-binding sites of the Texel sheep were one more in number than those of the Hu sheep (727), the rest of the RNA-binding regions were the same between the Hu sheep and Texel sheep, but with different positions (Figure 5).

SWISS-MODEL was used to analyze the tertiary structures of the ARNT from the Hu and Texel sheep, respectively. It was discovered through a comparison of the tertiary structures of the ARNT protein sequences from the Hu sheep and Texel sheep that the folding pattern of the ARNT protein in the Hu sheep, in terms of its tertiary structure, differed from that in the Texel sheep (Figure 6a). The tertiary structures of the ARNT protein in the Hu sheep and Texel sheep had the same number of β-folds, but the tertiary structure of the Hu sheep included one additional α-helix at amino acids 345–347, yielding 14 α-helices instead of 13 (Figure 6b). Additionally, the ARNT protein clustal structure in the Hu sheep differed from that of the Texel sheep by one protein clustal, consisting of five amino acids at amino acids 98–102. (Figure 6c). In addition, they were clustered with the crystal structure of the heterodimeric npas1-arnt complex (SMTL ID: 5sy5.4) [7], having 97.39% similarity. Furthermore, a phylogenetic tree was constructed for ARNT using the neighbor-joining method (Figure 7). The Hu sheep and goat (*Capra hircus*) first gathered into one branch and then gathered together with the sheep (*Ovis aries*).

## 4. Discussion

Research on the *ARNT* gene has mainly focused on its roles in mouse and human tumors and its molecular mechanisms [33,34,35,36]. Meanwhile, some research on the *ARNT* gene has focused on myofiber type determination and muscle regeneration [20,21], whereas research on its roles in sheep growth traits is relatively rare. As a result of our previous proteomic studies, we found that the ARNT protein and its potential functions might affect the growth and development of sheep embryonic muscle. However, the function and regulation mechanism of this gene have not been reported in regard to sheep growth traits. To reveal the association between the *ARNT* gene and sheep growth and development, we performed a quantitative analysis of the muscle tissues of different breeds of sheep. These Chinese Merino sheep were meat and wool breeds from Xinjiang, China, and the *ARNT* expression at D85 was significantly higher than at D135 (Figure 1a), and the degree of gene expression decreased gradually with muscle maturation, suggesting that a negative association may exist between the *ARNT* expression and muscle growth and development in Chinese Merino embryos. Furthermore, the *ARNT* gene is an essential component of HIF, and the different oxygen levels in sheep can affect the ability of the *ARNT* gene to regulate muscle growth and development. To explore whether there are differences in the *ARNT* gene function among sheep at different altitudes, the Hu sheep, a meat sheep breed that is famous in China, and the Gangba sheep, a special meat sheep breed from Tibet, were used to reveal the potential functions. At the level of the myoblasts, there was no significant difference between the Hu and Gangba sheep myoblasts (Figure 1b). Although the expression of the *ARNT* gene showed no significant difference in the afterborn Gangba sheep (Figure 1c), this expression of the gene showed a significant difference between the pre- and post-weaning afterborn Hu sheep (Figure 1d). These results confirm that the *ARNT* gene functions differently in sheep at different altitudes. However, this gene may be a key factor in the growth and development of sheep.

To date, there have been few studies analyzing the SNPs in the *ARNT* gene. To better understand the polymorphism and expression of the *ARNT* gene, our study analyzed the polymorphism of the *ARNT* gene in the Hu and Ujimqin sheep populations. The polymorphism analysis showed that the *ARNT* gene SNP loci in the Hu sheep were less numerous than in the Ujimqin sheep. In addition, there was only one SNP locus (rs413597480 A > G) in the Hu sheep. The genetic variation at this locus was small in the Hu sheep population. However, this locus had a favorable relationship with the Hu sheep growth traits. The BW at weaning was strongly associated with rs413597480 A > G. The rs413597480 A > G in the Hu sheep also had a strong association with the BFT at 5-month-old, as was the case for the CC and CW in the developmental stages of 5- and 6-month-old in the Hu sheep (*p* < 0.05). It is worth noting that the GG genotype was the dominant genotype at the rs413597480 A > G mutation locus in the Hu sheep, with substantial values for all the statistically linked growth characteristics, while the expression of the *ARNT* gene was significantly different at the weaning stage in the Hu sheep compared to other developmental stages (*p* < 0.01). These results demonstrated that the rs413597480 A > G of the *ARNT* gene has the potential to regulate growth and development during the weaning period and to affect the BFT of Hu sheep during the 5-month-old stage.

As for the *ARNT* gene in the Ujimqin sheep, there were five SNPs, of which the rs162298018 G > C SNP was a G/C transversion, and this SNP had a good statistical association with the BW and CC of 4-month-old sheep and CC and the TC of 6-month-old Ujimqin sheep, respectively. Furthermore, this SNP had high heterozygosity and a large amount of polymorphic information. In addition, there was strong linkage disequilibrium among rs162298018 G > C and rs159644025 G > A. According to this result, the rs162298018 G > C mutation was ruled out by chance. The remaining three SNP loci of the *ARNT* gene in the Ujimqin sheep were also associated with the growth traits, except for the rs427535938 G > A, while a haplotype domain was detected between rs159644025 G > A, rs421351865 G > A, rs401758103 A > G, and rs427535938 G > A, and the GGAG was the dominant haplotype. Based on the above results, both the analysis of the *ARNT* gene expression and the analysis of the genetic stability and the association of the mutant loci with the growth traits suggest that this gene may have some relevance to sheep growth and development, and this conclusion is similar to that of the studies of the influence of *ARNT* on the muscle in human and animal models [18,19,20].

Then, we further investigated the *ARNT* CDS sequence and protein. At present, there is no complete report of the *ARNT* gene CDS sequence in sheep. The Hu sheep is bred for meat in China, and there is no CDS cDNA sequence of the ARNT gene in Hu sheep. In this study, the complete CDS sequences of the Hu sheep *ARNT* gene were successfully obtained. The phylogenetic tree showed that the Hu sheep ARNT protein had the closest relationship with goat and sheep. The similarity was over 99%, indicating that the ARNT protein has been highly conserved throughout evolution. Additionally, the analysis of the CDS sequence showed that a G > C of the *ARNT* CDS sequence resulted in a nonconservative missense point mutation, leading to a change from proline to alanine at position 1948 bp. Notably, the non-conserved missense point mutation in the *ARNT* CDS sequence that we cloned from the Hu sheep was the same single-base mutation as the rs162298018 G > C in the Ujimqin sheep, and the GG genotype of this mutation site was the dominant genotype that was significantly associated with some growth traits in the Hu and Ujimqin sheep. The Hu sheep and Ujimqin sheep with this genotype had greater body weight and body size traits. These findings not only confirm the ability of the *ARNT* gene to control growth and development in sheep, but they also raise the possibility that the G > C mutation at 1948 bp in the CDS region may be a significant candidate SNP influencing the growth and development of Hu and Ujimqin sheep.

Meanwhile, the secondary and tertiary structure analysis of the ARNT protein of the Hu sheep revealed that this nonconservative missense has significant influences on the secondary and tertiary structures of ARNT. Compared to the Texel sheep, there are more DNA- and protein-binding sites in the ARNT protein in the Hu sheep. Some researchers reported that specific SNPs can affect the secondary structures of mRNAs and proteins [37,38], indicating that the DNA- or protein-binding sites of the Hu sheep ARNT protein may play a crucial role in gene function and that this can change the gene function in regard to sheep growth and development [39,40,41,42]. While the changes in the tertiary conformation of the ARNT protein and the increase in the α-helix after the G > C mutation suggest that the G > C mutation at 1948 bp in the CDS region of the *ARNT* gene may affect the function of the ARNT protein and, thus, the regulation of growth traits in Hu and Ujimqin sheep, they also suggest that the G > C mutation at 1948 bp in the CDS region of the *ARNT* gene may be a potential molecular regulatory locus for the growth traits of sheep.

Based on the results of our analysis, we suggest that the *ARNT* gene, with its key potential function, regulates the growth traits of sheep. The G > C mutation at 1948 bp in the cloned *ARNT* CDS sequence of the Hu sheep was the same locus mutation as rs162298018 G > C in the Ujimqin sheep, which resulted in a nonconservative missense point mutation, leading to a change from proline to alanine, and this SNP locus was significantly associated with the growth traits of the Ujimqin sheep. Meanwhile, we also found that the results of the ARNT protein function analysis are consistent with the quantitative results for the *ARNT* gene in the Hu sheep muscle tissues, indicating that the *ARNT* gene may play an important functional role in affecting the muscle growth and development of Hu and Ujimqin sheep. At present, we do not know the specific function of the DNA- or protein-binding regions in the ARNT protein. However, it can be seen that the growth traits are associated with the functionality of *ARNT* in sheep. In the future, we will perform more functional analyses to clarify our doubts. For example, we will create an overexpression vector containing the rs162298018 G > C mutation site of *ARNT*. We will also investigate the changes in the *ARNT* gene function before and after the mutation, as well as their effects on myoblast growth and development. Additionally, the key factors that bind to DNA- and protein-binding sites will also be predicted.

## 5. Conclusions

In this study, we found that the *ARNT* gene is significantly expressed in Chinese merino embryos and the muscle tissues of Hu sheep after birth. Additionally, we successfully screened six SNPs in *ARNT* and genotyped these loci. The linkage disequilibrium analysis showed that there is a strong linkage disequilibrium between rs162298018 G > C and rs159644025 G > A, and the association analysis showed a significant association between rs413597480 A > G and the growth traits of Hu sheep (*p* < 0.05), and four SNP loci (rs162298018 G > C, rs159644025 G > A, rs421351865 G > A, and rs401758103 A > G) were associated with growth traits of Ujimqin sheep (*p* < 0.05). Meanwhile, we discovered that the G > C mutation at 1948 bp in the cloned *ARNT* CDS sequence of Hu sheep is the same as the rs162298018 G > C in Ujimqin sheep. A bioinformatic analysis revealed that this G > C variant caused a nonconservative missense point mutation, resulting in a change from proline to alanine. This nonconservative missense mutation not only influences the function of the ARNT secondary protein structure but also affects the tertiary protein structure. For the first time, this study systematically analyzed the expression, structure, and function of the *ARNT* gene and its encoded proteins in Hu sheep, and we proposed that the G > C mutation at 1948 bp in the CDS region of the *ARNT* gene may be a possible molecular regulatory marker of sheep growth traits. This provided a basis for further research aiming to unravel the functional mechanisms of the *ARNT* gene in sheep and the screening of key molecular breeding markers.

## Figures and Tables

**Figure 1 biology-11-01795-f001:**
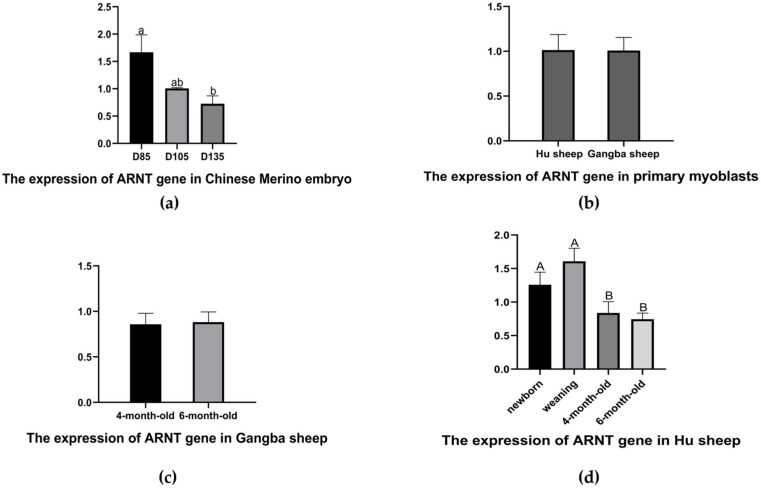
*ARNT* gene expression in muscle tissue of different breeds of sheep and primary myoblasts: (**a**) Chinese Merino sheep embryonic muscle tissues; (**b**) the expression of *ARNT* in Hu and Gangba sheep primary myoblasts; (**c**) The expression of *ARNT* in Gangba sheep muscle tissues; (**d**) The expression of *ARNT* in Hu sheep muscle tissues. Note: a and b indicate a significant difference (*p* < 0.05), and A and B indicate an extremely significant difference (*p* < 0.01).

**Figure 2 biology-11-01795-f002:**
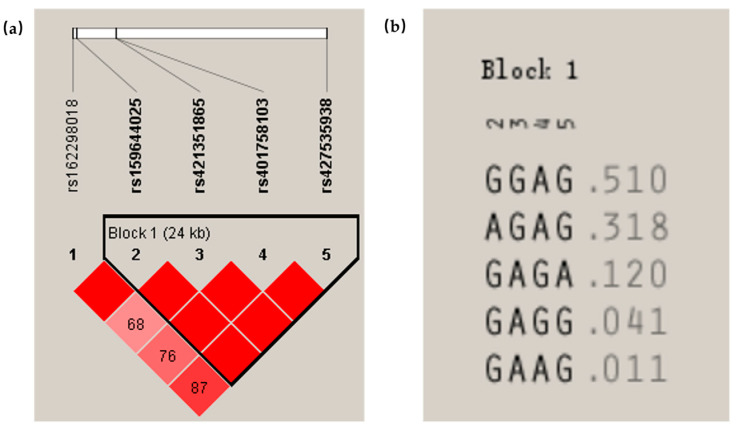
Analysis of the linkage disequilibrium of five SNPs of the *ARNT* gene: (**a**) Red squares indicate high levels of interlocking disequilibrium, D’ = 1 indicates complete interlocking disequilibrium, D’ = 0 indicates complete interlocking equilibrium, D values between SNPs in the squares where they cross are shown, no number indicates a D value of 1, and 1–5 correspond to the selected SNPs in Table 1. (**b**) One haploid domain was detected.

**Figure 3 biology-11-01795-f003:**
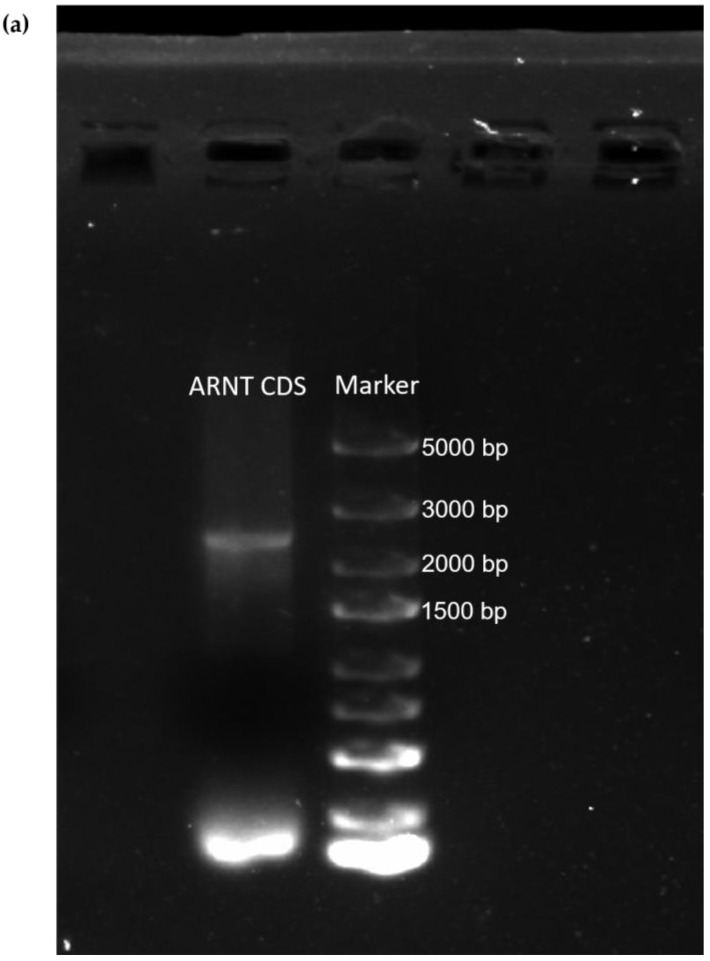
*ARNT* gene CDS sequence in Hu and Texel sheep: (**a**) Electrophoresis agarose gel of Hu sheep *ARNT* CDS PCR product, (**b**) CDS alignment of *ARNT* gene between the Hu sheep and Texel sheep. The top sequence is the Hu sheep *ARNT* CDS sequence, and the bottom sequence is the Texel sheep *ARNT* CDS sequence, and the two SNP mutations are in the red box, respectively. And add one * per 10 bases in Figure 3b.

**Figure 4 biology-11-01795-f004:**
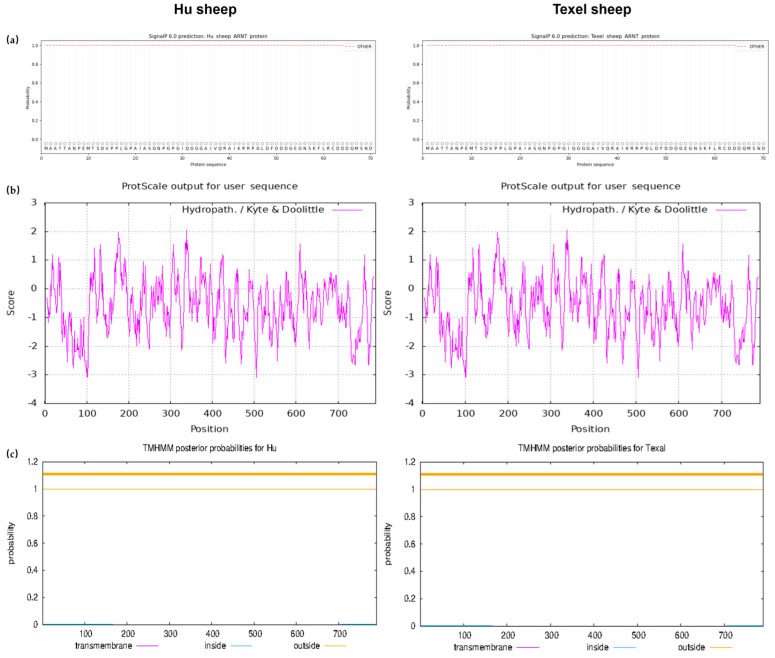
ARNT protein structure and property prediction: (**a**) Signal peptide prediction in Hu and Texel sheep, (**b**) Hydrophilic and hydrophobic prediction, (**c**) Prediction of transmembrane helices.

**Figure 5 biology-11-01795-f005:**
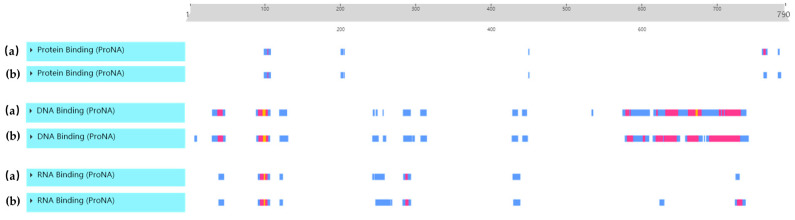
ARNT secondary structures of (a) Hu and (b) Texel sheep. The blue box refers to low binding sites (RI:0–33); the red box refers to intermediate binding sites (RI:34–66); and the yellow box refers to high binding sites (RI:66–100).

**Figure 6 biology-11-01795-f006:**
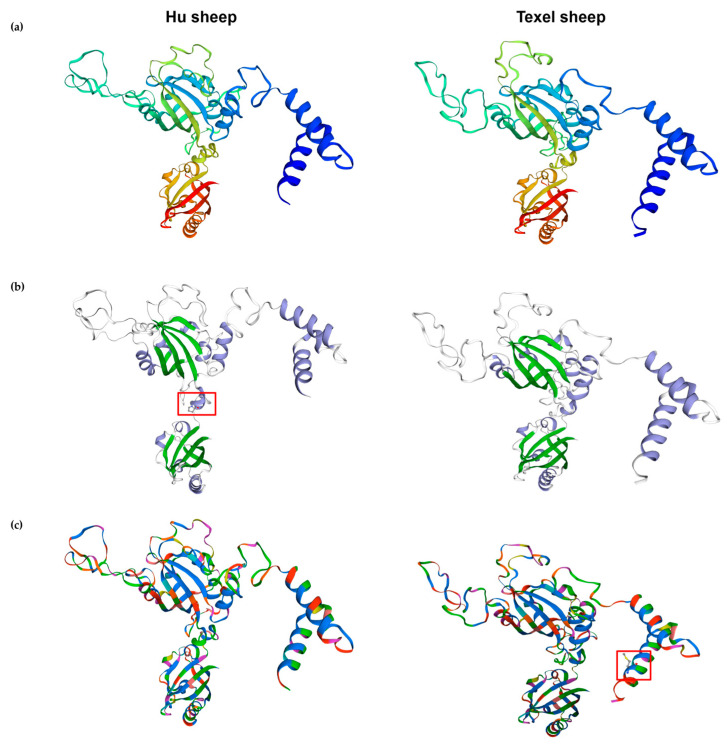
ARNT tertiary structure prediction and analysis: (**a**) ARNT tertiary structure in Hu and Texel sheep; (**b**) β-fold and α-helix of ARNT protein. The green peptide is β-fold, the purple peptide is α-helix, the red box shows one more α-helix in the Hu sheep than Texel sheep. (**c**) ARNT protein clustal of the Hu and Texel sheep. The red box shows one more protein clustal in the Hu sheep than Texel sheep.

**Figure 7 biology-11-01795-f007:**
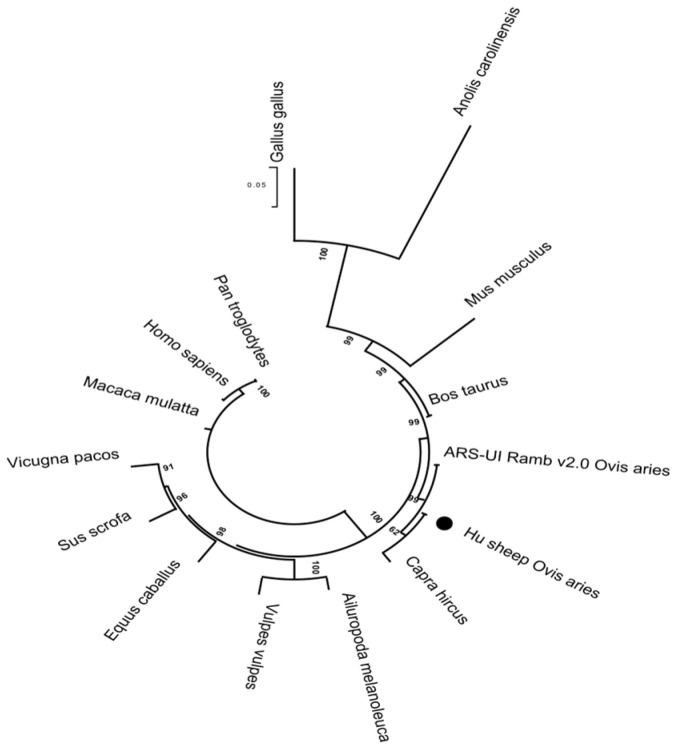
Phylogenetic analysis of the ARNT protein.

**Table 1 biology-11-01795-t001:** SNPs observed in the *ARNT* gene.

Gene Name	SNP Mutation Site	Nucleotide Change	Consequence Type
*ARNT*	rs413597480	A > G	intron variant
	rs162298018	G > C	exon variant
	rs159644025	G > A	intron variant
	rs421351865	G > A	intron variant
	rs401758103	A > G	intron variant
	rs423213327	NA	NA
	rs404936302	NA	NA
	rs426270191	NA	NA
	rs427535938	G > A	intron variant
	rs407279804	NA	NA

**Table 2 biology-11-01795-t002:** Genetic parameters of the single-nucleotide polymorphism (SNP) markers of the *ARNT* gene in the Hu and Ujimqin sheep.

SNP Locus	Breed	Genotype	Number	Genotypic Frequency	Allele	Allelic Frequency	Ho	Ne	He	PIC	HWE Test (*p*-Value)
rs413597480	Hu sheep	AA	2259	0.76	A	0.50	0.78	0.22	1.29	0.20	0.67
		AG	675	0.23	G	0.50					
		GG	47	0.02							
rs162298018	Ujimqin sheep	CC	81	0.24	C	0.50	0.50	0.50	2.00	0.37	0.58
		GC	171	0.52	G	0.50					
		GG	80	0.24							
rs159644025	Ujimqin sheep	AA	35	0.11	A	0.32	0.57	0.43	1.77	0.34	0.75
		GA	142	0.43	G	0.68					
		GG	156	0.47							
rs421351865	Ujimqin sheep	AA	11	0.03	A	0.17	0.72	0.28	1.39	0.24	0.54
		AG	89	0.27	G	0.83					
		GG	227	0.69							
rs401758103	Ujimqin sheep	AA	237	0.71	A	0.84	0.73	0.27	1.37	0.23	0.33
		AG	85	0.26	G	0.16					
		GG	11	0.03							
rs427535938	Ujimqin sheep	AA	5	0.02	A	0.12	0.79	0.21	1.27	0.19	0.92
		AG	70	0.21	G	0.88					
		GG	258	0.77							

**Table 3 biology-11-01795-t003:** Association analysis between the *ARNT* gene polymorphisms and growth traits of Hu sheep from birth to 2-month-old.

Loci	Genotype (Number)	Birth Body Weight/kg	Weaning Weight/kg	Body Weight at 2-Month-Old/kg
rs413597480 A > G	AA (n = 1476)	4.05 ± 0.02	14.86 ± 0.07 ^a,b^	19.9 ± 0.09
	GA (n = 456)	4.06 ± 0.03	14.68 ±0 .13 ^b^	19.84 ± 0.17
	GG (n = 42)	4.16 ± 0.12	15.59 ± 0.47 ^a^	20.74 ± 0.63

Note: phenotypic data in the table are presented as mean ± SE. ^a^ and ^b^ indicate significant differences.

**Table 4 biology-11-01795-t004:** Association analysis between the *ARNT* gene polymorphisms and growth traits (BW, BH, BL, CC, and TC) of Hu sheep from 3-month-old to 6-month-old.

Loci	Age	Genotypes	BW/kg	BH/cm	BL/cm	CC/cm	TC/cm
		(Number)					
rs413597480 A > G	3-month- old	AA (n = 700)	28.24±0.17	58.71 ± 0.09	62.39 ± 0.11	70.43 ± 0.17	7.63 ± 0.02
		GA (n = 230)	28.33 ± 0.3	58.75 ± 0.16	62.42 ± 0.2	70.76 ± 0.3	7.69 ± 0.04
		GG (n = 20)	29.12 ± 0.77	59.33 ± 0.42	62.5 ± 0.61	72.16 ± 0.93	7.63 ± 0.11
	4-month- old	AA (n = 752)	35.76 ± 0.19	61.4 ± 0.08	66.62 ± 0.11	75.98 ± 0.15	8.17 ± 0.02
		GA (n = 240)	36.09 ± 0.33	61.51 ± 0.15	66.79 ± 0.18	75.89 ± 0.27	8.15 ± 0.04
		GG (n = 22)	36.53 ± 0.9	61.51 ± 0.42	66.9 ± 0.64	76.55 ± 0.66	8.14 ± 0.11
	5-month- old	AA (n = 741)	42.67 ± 0.21	64.21 ± 0.09	71.53 ± 0.11	81.11 ± 0.16 ^b^	8.55 ± 0.02
		GA (n = 238)	43 ± 0.38	64.48 ± 0.16	71.54 ± 0.18	81.05 ± 0.29 ^b^	8.51 ± 0.03
		GG (n = 22)	43.96 ± 1.08	64.38 ± 0.33	71.11 ± 0.5	82.82 ± 0.75 ^a^	8.54 ± 0.11
	6-month-old	AA (n = 529)	48.62 ± 0.28	65.78 ± 0.11	75.33 ± 0.12	85.95 ± 0.21	8.89 ± 0.03
		GA (n = 169)	49.27 ± 0.48	66.15 ± 0.18	75.3 ± 0.2	86.52 ± 0.35	8.97 ± 0.05
		GG (n = 16)	50.58 ± 1.3	66.16 ± 0.52	75.07 ± 0.79	87.29 ± 1.11	9.02 ± 0.08

Note: body weight (BW), body height (BH), body length (BL), chest circumference (CC), tube circumference (TC). Phenotypic data in the table are presented as mean ± SE. ^a^ and ^b^ indicate significant differences.

**Table 5 biology-11-01795-t005:** Association analysis between the *ARNT* gene polymorphisms and growth traits (CW, CD, HW, CH, BFT, and EMA) of Hu sheep from 3-month-old to 6-month-old.

Loci	Age	Genotypes	CW/cm	CD/cm	HW/cm	CH/cm	BFT/mm	EMA/cm^2^
		(Number)						
rs413597480 A > G	3-month-old	AA (n = 700)	13.52 ± 0.04	22.18 ± 0.05	13.44 ± 0.04	60 ± 0.1	NA	NA
		GA (n = 230)	13.6 ± 0.08	22.24 ± 0.11	13.54 ± 0.08	60.08 ± 0.18	NA	NA
		GG (n = 20)	13.69 ± 0.19	22.74 ± 0.32	13.76 ± 0.19	60.83 ± 0.47	NA	NA
	4-month-old	AA (n = 752)	13.83 ± 0.05	24.65 ± 0.06	13.54 ± 0.04	63.09 ± 0.1	NA	NA
		GA (n = 240)	13.98 ± 0.09	24.66 ± 0.11	13.63 ± 0.07	63.26 ± 0.17	NA	NA
		GG (n = 22)	14.29 ± 0.29	25.05 ± 0.28	13.74 ± 0.2	63.56 ± 0.47	NA	NA
	5-month-old	AA (n = 741)	16.48 ± 0.05	25.22 ± 0.06	NA	NA	7.02 ± 0.04 ^b^	8.13 ± 0.04
		GA (n = 238)	16.63 ± 0.09	25.31 ± 0.11	NA	NA	7.14 ± 0.07 ^b^	8.17 ± 0.07
		GG (n = 22)	16.36 ± 0.26	25.33 ± 0.26	NA	NA	7.67 ± 0.28 ^a^	8.37 ± 0.23
	6-month-old	AA (n = 529)	18.37 ± 0.07 ^b^	26.62 ± 0.08	NA	NA	7.53 ± 0.06	9.59 ± 0.06
		GA (n = 169)	18.65 ± 0.14 ^a,b^	26.86 ± 0.14	NA	NA	7.78 ± 0.1	9.79 ± 0.09
		GG (n = 16)	19.1 ± 0.34 ^a^	27.14 ± 0.35	NA	NA	7.46 ± 0.25	9.85 ± 0.28

Note: chest width (CW), chest depth (CD), backfat thickness (BFT), eye muscle area (EMA), hip width (HW), cruciate section height (CH). Phenotypic data in the table are presented as mean ± SE. ^a^ and ^b^ indicate significant differences.

**Table 6 biology-11-01795-t006:** Association analysis between the *ARNT* gene polymorphisms and growth traits of 4-month-old Ujimiqin sheep.

Loci	Genotypes (Number)	BW/kg	BH/cm	BL/cm	CC/cm	TC/cm
rs162298018 G > C	CC (n = 78)	29 ± 0.63 ^a,b^	59.06 ± 0.3	61.94 ± 0.26	76.97 ± 0.69 ^a,b^	7.76 ± 0.05
	GC (n = 165)	28.58 ± 0.43 ^b^	59.21 ± 0.18	61.56 ± 0.19	76.06 ± 0.47 ^b^	7.74 ± 0.04
	GG (n = 75)	29.6 ± 0.59 ^a^	59.45 ± 0.32	62.07 ± 0.3	77.33 ± 0.66 ^a^	7.81 ± 0.05
rs159644025 G > A	AA (n = 35)	29.06 ± 1	59.27 ± 0.48	61.9 ± 0.42	77.4 ± 1.14	7.79 ± 0.09
	AG (n=135)	28.49 ± 0.47	59.04 ± 0.2	61.48 ± 0.19	76.28 ± 0.53	7.73 ± 0.04
	GG (n=149)	29.28 ± 0.43	59.42 ± 0.22	62.03 ± 0.22	76.63 ± 0.46	7.79 ± 0.04
rs401758103 A > G	AA (n = 10)	31.19 ± 1.51 ^a^	59.7 ± 0.59	62.9 ± 0.41	77.15 ± 1.28	7.85 ± 0.11
	AG (n = 87)	28.82 ± 0.59 ^b^	58.97 ± 0.29	61.84 ± 0.28	76.19 ± 0.63	7.72 ± 0.05
	GG (n = 217)	28.76 ± 0.37 ^b^	59.3 ± 0.17	61.69 ± 0.16	76.56 ± 0.42	7.77 ± 0.03
rs421351865 G > A	AA (n = 225)	28.9 ± 0.36 ^b^	59.34 ± 0.17	61.74 ± 0.16	76.65 ± 0.41	7.76 ± 0.03
	AG (n = 84)	28.71 ± 0.59 ^b^	58.93 ± 0.3	61.75 ± 0.28	76.29 ± 0.63	7.75 ± 0.06
	GG (n = 10)	31.19 ± 1.51 ^a^	59.7 ± 0.59	62.9 ± 0.41	77.15 ± 1.28	7.85 ± 0.11
rs427535938 G > A	AA (n = 5)	30.08 ± 2.11	58.8 ± 1.01	63.2 ± 0.58	75.8 ± 1.85	7.7 ± 0.12
	AG (n = 68)	28.52 ± 0.67	59.04 ± 0.29	61.44 ± 0.29	76.1 ± 0.71	7.75 ± 0.06
	GG (n = 246)	29.01 ± 0.34	59.3 ± 0.16	61.85 ± 0.16	76.71 ± 0.39	7.77 ± 0.03

Note: Body weight (BW), Body height (BH), Body length (BL), Chest circumference (CC), Tube circumference (TC). Phenotypic data in the table are presented as Mean ± S.E. ^a^ and ^b^ indicates significant difference.

**Table 7 biology-11-01795-t007:** Association analysis between the *ARNT* gene polymorphisms and growth traits of 6-month-old Ujimiqin sheep.

Loci	Genotypes (Number)	BW/kg	BH/cm	BL/cm	CC/cm	TC/cm	CW/cm	CD/cm
rs162298018 G > C	CC (n = 64)	33.39 ± 0.75	60.47 ± 0.41	64.5 ± 0.42	83.14 ± 0.86 ^a^	7.59 ± 0.06 ^a,b^	15.16 ± 0.27	31.36 ± 0.41
	GC (n = 143)	32.17 ± 0.45	59.94 ± 0.27	64.07 ± 0.36	80.94 ± 0.53 ^b^	7.52 ± 0.03 ^b^	14.66 ± 0.19	30.84 ± 0.22
	GG (n = 67)	32.76 ± 0.69	60.51 ± 0.37	64.48 ± 0.44	82.22 ± 0.73 ^a,b^	7.66 ± 0.05 ^a^	14.99 ± 0.29	31.52 ± 0.28
rs159644025 G > A	AA (n = 29)	34.4 ± 1.28 ^a^	61.07 ± 0.64 ^a^	65.24 ± 0.56 ^a^	84.07 ± 1.56 ^a^	7.62 ± 0.1	15.59 ± 0.44 ^a^	30.83 ± 0.75
	AG (n = 115)	31.94 ± 0.5 ^b^	59.9 ± 0.29 ^b^	63.74 ± 0.42 ^b^	80.86 ± 0.59 ^b^	7.49 ± 0.04	14.73 ± 0.2 ^b^	30.84 ± 0.25
	GG (n = 131)	32.78 ± 0.46 ^b^	60.3 ± 0.27 ^a,b^	64.53 ± 0.31 ^a,b^	82.05 ± 0.51 ^b^	7.62 ± 0.04	14.8 ± 0.21 ^b^	31.46 ± 0.21
rs401758103 A > G	AA (n = 10)	34.03 ± 1.48	59.6 ± 1.08	64.6 ± 1.27	81.7 ± 1.37	7.79 ± 0.12 ^a^	15.7 ± 0.65	31.7 ± 0.47
	AG (n = 69)	32.81 ± 0.67	60.16 ± 0.36	64.19 ± 0.39	82.77 ± 0.74	7.56 ± 0.05 ^b^	14.52 ± 0.25	31.62 ± 0.34
	GG (n = 190)	32.47 ± 0.41	60.28 ± 0.23	64.2 ± 0.3	81.31 ± 0.48	7.56 ± 0.03 ^b^	14.87 ± 0.16	30.91 ± 0.2
rs421351865 G > A	AA (n = 197)	32.51 ± 0.4	60.27 ± 0.23	64.26 ± 0.29	81.48 ± 0.47	7.57 ± 0.03 ^b^	14.88 ± 0.16	30.96 ± 0.2
	AG (n = 68)	32.66 ± 0.67	60.12 ± 0.38	64.29 ± 0.43	82.62 ± 0.74	7.55 ± 0.05 ^b^	14.65 ± 0.29	31.54 ± 0.33
	GG (n = 10)	34.03 ± 1.48	59.6 ± 1.08	64.6 ± 1.27	81.7 ± 1.37	7.79 ± 0.12 ^a^	15.7 ± 0.65	31.7 ± 0.47
rs427535938 G > A	AA (n = 5)	32.36 ± 2.08	58.8 ± 2.03	63.2 ± 1.8	81.4 ± 2.29	7.72 ± 0.18	15.4 ± 0.75	31.6 ± 0.75
	AG (n = 5)	32.43 ± 0.73	60.25 ± 0.37	64.07 ± 0.44	82.25 ± 0.8	7.55 ± 0.05	14.58 ± 0.31	31.69 ± 0.35
	GG (n = 215)	32.65 ± 0.39	60.23 ± 0.22	64.36 ± 0.28	81.65 ± 0.45	7.57 ± 0.03	14.91 ± 0.16	30.98 ± 0.19

Note: body weight (BW), body height (BH), body length (BL), chest circumference (CC), tube circumference (TC), chest width (CW), chest depth (CD). Phenotypic data in the table are presented as mean ± SE. ^a^ and ^b^ indicate significant differences.

## Data Availability

Data are available on request.

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
