# Peer review of "Polymorphism, Expression, and Structure Analysis of a Key Gene ARNT in Sheep (*Ovis aries*)"

_biology, 2022, doi:10.3390/biology11121795_

Round 1

Reviewer 1 Report

This manuscript investigated Polymorphism, expression, and structure analysis of a key gene  ARNT in sheep (Ovis aries). The content is fall into the scope of the present journal. The topic is interest, the experiment was well conducted, and the manuscript also raised many concerns. The follow are some specific comments.

Are you tested the normal distribution of the phenotypic data related to the considered growth traits

Lines 130 -135. Preferably placed under a separate heading (Statistical analysis)

Line 165….What is the methodology that you used to estimate the genetic parameters…. Please clarify.

Line 170….The other fixed factors should be incorporated in the statistical model such as birth weight, birth type, and birth season as well as dam parity as classes, etc… not just animal gender as you referred in order to adjust for environmental effects. I suggest that the effect of sire should be studied as a random effect to determine the breeding values.

Figure 1.. Please revise the significant differences between newborn and weaning columns in Figure 1-D. It seems to be non-significant due to the high variations as showed on the standard bar and you confirmed significant differences.

You must insert a table to show the overall average of phenotypic data and then link that with the ARNT gene polymorphism.

Author Response

Response to Reviewer 1 Comments

Point 1: Are you tested the normal distribution of the phenotypic data related to the considered growth traits

Response 1: Thank you very much for your suggestion, we tested the normal distribution of phenotypic data related to the growth traits under consideration, as detailed in lines 185-187 of this paper.

Point 2: Lines 130 -135. Preferably placed under a separate heading (Statistical analysis)

Response 2: We have made detailed revisions based on your comments, as detailed in lines 136-141 of this paper.

Point 3: Line 165….What is the methodology that you used to estimate the genetic parameters…. Please clarify.

Response 3: We have described in detail the methodology of the genetic parameters, as detailed in lines 157-161.

Point 4: Line 170…The other fixed factors should be incorporated in the statistical model such as birth weight, birth type, and birth season as well as dam parity as classes, etc… not just animal gender as you referred in order to adjust for environmental effects. I suggest that the effect of sire should be studied as a random effect to determine the breeding values.

Response 4: We have refined the general linear model and done a new association analysis. However, we do not have full sire information for the phenotypic data to include the sire effect in the model, as detailed in lines 184-195 and Table 3, 4a and 4b. I have benefited from your suggestions, and we will implement them in the future. Thank you very much.

Point 5: Figure 1. Please revise the significant differences between newborn and weaning columns in Figure 1-D. It seems to be non-significant due to the high variations as showed on the standard bar and you confirmed significant differences.

Response 5: The ARNT gene expression significance has been revised according to the latest results, as detailed in lines 256-258 and figure 1.

Point 6: You must insert a table to show the overall average of phenotypic data and then link that with the ARNT gene polymorphism.

Response 6: We have carefully modified the table of results of association analysis between ARNT SNP loci and phenotypic traits as suggested, as detailed in tables 3-6 and lines 314-358.

Reviewer 2 Report

Nicely written paper on ARNT gene in sheep. Here are some suggestions to improve the paper:

* What was the rationale behind choosing particular months old sheep for this study (line 111-112)

* What is the power of this study?

* Could you explain the linkage disequilibrium of associated markers in ARNT genes (Table 1)

* You have mentioned about more functional analysis is needed for future research. Can you please elaborate it on this paper?

Author Response

Response to Reviewer 2 Comments

Point 1: What was the rationale behind choosing particular months old sheep for this study (line 111-112)

Response 1: For Chinese Merino sheep, specific times were selected for muscle tissue collection. This is because we chose critical time points for embryonic muscle growth and development. The samples used for our proteomics in the earlier period were the same as those used in this paper, as described in detail in lines 110–113. Meanwhile, newborn, weaning, 4, and 6-month-old are the key points of growth and development of sheep after birth, so we selected the above time points of Hu and Gangba sheep muscle samples for the study.

Point 2: What is the power of this study?

Response 2: In order to mine key candidate genes regulating growth traits in sheep, we previously analyzed Chinese Merino sheep embryonic muscle tissue using TMT proteomic method. We identified ARNT as a possible key candidate gene affecting sheep embryonic muscle growth and development, but the function of the ARNT gene was not studied and validated. Based on the above findings, this study was conducted to verify the relationship between the ARNT gene and sheep growth traits at the molecular and population levels. By conducting ARNT protein studies, it was also possible to analyze the potential functions of ARNT in affecting sheep growth and development. This research goal was to provide a reference for screening molecular genetic markers related to sheep growth traits and analyzing regulatory mechanisms.

Point 3: Could you explain the linkage disequilibrium of associated markers in ARNT genes (Table 1)

Response 3:We have performed a linkage disequilibrium analysis of the ARNT gene SNP loci based on your comments, further refining the rigor and systematicity of this study, as detailed in lines 299-312.

Point 4: You have mentioned about more functional analysis is needed for future research. Can you please elaborate it on this paper?

Response 4: We have provided a brief description of possible future research as detailed on lines 547-551.

Round 2

Reviewer 1 Report

Thank you very much. All the modifications have been made and the manuscript has become suitable for publication in the Biology Journal. Just:

1- please mention the name of the normal test that you performed

2- Please replace the title of Paragraph 2.4.2 with Statistical Analysis  instead of association analysis

3- Please revise a formentioned paragraphe as there are several mistakes; + should be repleaced by ±

4- Please add the reference of R program

Author Response

Point 1: please mention the name of the normal test that you performed.

Response 1: Thank you very much for your suggestion. The name of the normal analysis has been added, see lines 186-188 for details.

Point 2: Please replace the title of Paragraph 2.4.2 with Statistical Analysis instead of association analysis

Response 2: The title of Paragraph 2.4.2 has been changed to Statistical Analysis, detailed in line 183.

Point 3: Please revise a formentioned paragraphe as there are several mistakes; + should be repleaced by ±

Response 3: The error you pointed out has been corrected, detailed in lines 190-191.

Point 4: Please add the reference of R program

Response 4: The phenotypic data correction was performed in R language using GLM, relevant reference was added in line 186, and the detailed running code was shown in Table S2.
